# Synchronization between Attractors: Genomic Mechanism of Cell-Fate Change

**DOI:** 10.3390/ijms241411603

**Published:** 2023-07-18

**Authors:** Masa Tsuchiya, Paul Brazhnik, Mariano Bizzarri, Alessandro Giuliani

**Affiliations:** 1SEIKO Life Science Laboratory, SEIKO Research Institute for Education, Osaka 540-6591, Japan; 2Academy of Integrated Science, Virginia Tech, Blacksburg, VA 24061, USA; brazhnik@vt.edu; 3Systems Biology Group, Department of Experimental Medicine, University La Sapienza, 00163 Roma, Italy; mariano.bizzarri@uniroma1.it; 4Environment and Health Department, Istituto Superiore di Sanitá, 00161 Rome, Italy

**Keywords:** genome expression, cell-fate decision, reversion of cancer, self-organized criticality (SOC), critical point, genome engine, genome attractor, biological regulation, transition theory

## Abstract

Herein, we provide a brief overview of complex systems theory approaches to investigate the genomic mechanism of cell-fate changes. Cell trajectories across the epigenetic landscape, whether in development, environmental responses, or disease progression, are controlled by extensively coordinated genome-wide gene expression changes. The elucidation of the mechanisms underlying these coherent expression changes is of fundamental importance in cell biology and for paving the road to new therapeutic approaches. In previous studies, we pointed at dynamic criticality as a plausible characteristic of genome-wide transition dynamics guiding cell fate. Whole-genome expression develops an engine-like organization (genome engine) in order to establish an autonomous dynamical system, capable of both homeostasis and transition behaviors. A critical set of genes behaves as a critical point (CP) that serves as the organizing center of cell-fate change. When the system is pushed away from homeostasis, the state change that occurs at the CP makes local perturbation spread over the genome, demonstrating self-organized critical (SOC) control of genome expression. Oscillating-Mode genes (which normally keep genome expression on pace with microenvironment fluctuations), when in the presence of an effective perturbative stimulus, drive the dynamics of synchronization, and thus guide the cell-fate transition.

## 1. Introduction

The actions of a few master regulators of gene expression [1] have been shown to guide cell-fate changes, where a cascade of events, starting from a small number of transcription factors, either induces or suppresses the coordinated expression of thousands of genes [2]. This global coordination suggests that major modifications in cell function at different organizational layers drive cell-fate transitions. However, the mechanistic interpretation of these changes as simple straightforward outcomes of gene–gene interactions must be reconciled with biological constraints, like the complex architecture of cell genome organization, the lack of a sufficient number of regulatory molecules [3,4,5], and microenvironment-dependent factors [6]. Therefore, in order to properly appreciate this complexity, we should apply additional organizational concepts toward understanding the cell-fate transition process from a systems biology perspective. In order to elucidate the mechanism of how the transition to a different global mode of expression occurs at the whole-genome scale, we consider genome expression as an integrated dynamical system, and not as the result of coarse graining on a multiplicity of local regulations; we also further exploit basic principles of dynamical systems theory and statistical mechanics. In non-equilibrium statistical mechanics, Self-Organized Criticality (classical SOC: c-SOC) was proposed as a general theory with which to accommodate self-organization and emergent order in thermodynamically open systems. Through considerable research over the past several decades, c-SOC has become an important topic in many scientific research domains, such as brain research and social science [7,8]. In genomics, our own findings [9,10,11,12,13,14,15,16] revealed that self-organized criticality (SOC) is also a plausible, physically motivated, candidate framework for massive, coordinated gene expression regulation.

The occurrence of SOC in cell-fate changes has been demonstrated using a mean-field approach that exploits the average characteristics of groups of genes instead of dealing with noisy expressions of individual genes. This technique is grounded in the existence of the Coherent Stochastic Behavior (CSB; see more in [9,10,11,12,13,14,15,16]) of gene expressions within sufficiently large gene groups. While individual genes within a given group display stochasticity, the group’s average expressions (centers of mass, CMs) quickly converge to certain values after the size of the group becomes sufficiently large. The genomic mechanism of cell-fate transitions has been verified at both the cell-population and single-cell levels, including those of HRG-Stimulated MCF-7 cells [17,18], atRA- and DMSO-stimulated HL-60 cells [19], Th17 cell differentiation from Th0 cells [20], and mouse and human early embryonic development [21,22].

We demonstrated that whole-genome expression is dynamically self-organized, through emergent critical behaviors (dynamic criticality), into a few distinct expression domains (critical states). The coexistence of SOC critical states has been confirmed through multivariate analysis of gene expression profiles [14].

A specific set of genes (grouped within the “critical point”, CP), displays a singular behavior. The CP emerges at the boundary between critical states and promotes the spread of the perturbation in gene expression across the entire genome. At critical transition, an abrupt change in the expression level of CP genes provokes a sort of ‘genome avalanche’ that, through a domino effect, affects the entire gene expression pattern. However, the SOC approach we adopted significantly differs from the classical one (c-SOC), which considers the phase transition from one global critical state to another (i.e., subcritical to supercritical genome state transition) as the genome is approaching a critical point (attractor). The set of genes forming the critical point (CP) in our SOC models initiates critical behavior, and the CP state change guides a global transition in the genome, whereas the CP in c-SOC corresponds to a specific state of the genome, and thus provides fundamentally different findings from those of our SOC. Moreover, at odds with c-SOC, the different states are contemporaneously present in the genome system in our SOC approach.

Below, we list several essential key points that are important for understanding the self-organized critical (SOC) control of genome expression in cell-fate transitions (see more in [13,15].

(1)The self-organization of genome expression becomes evident by means of *nrmsf* (Normalized Root Mean Square Fluctuation) a metric parameter quantifying the genes’ temporal variability, which uncovers distinct response domains (critical states).(2)Coherent Stochastic Behavior (CSB) occurs within sufficiently large gene groups. This is a consequence of the law of large numbers, which allows for the regularization of the noise present at the single-gene level. This implies that the center of mass (CM) of an ensemble of stochastic gene expressions acts as an attractor: the CMs of whole-genome expression and the critical states act as the genome attractor (GA) and local critical state attractors, respectively.(3)The autonomous control of fluxes between critical-state attractors gives rise to an open thermodynamic engine-like system (genome engine).(4)A specific set of critical genes, identified by the metric parameter (*nmrsf*), behaves as a critical point (CP). Genes within the CP display extremely high temporal variability and transmit their fluctuations across the genome. The state change in the CP gives rise to an expression change wave spreading across the entire genome that allows cell-fate change.(5)The initiation of genomic transition results in a synchronization between the CP and GA specifying the cell fate.(6)At the tipping point (critical transition), the switching of cyclic flow within the genome engine coherently enhances or suppresses the genome engine by revealing fluctuations in critical states.(7)The aforementioned dynamical perspective acquires a material basis in terms of Peri-centromeric Domains (PADs) bursting at the transition [14,23].

Here, we show how the synchronization dynamics between the CP and GA are modulated by Oscillating-Mode (OM) genes to guide the cell-fate critical transition.

At any layer of biological organizational hierarchy, from protein molecules to ecosystems, continuous oscillations occur within the system configuration around its typical state (be it a native protein structure or an abundance of species in an ecological system). These oscillations are due to the system’s responses to continuous challenges from environmental variations [24]. The adaptation to these challenges, in order to maintain global homeostasis (i.e., the resilience of the system’s attractor), is the task of so-called Oscillating-Mode (OM) genes, which display elevated temporal variance, mirroring such an endless adaptive process. The identification of such OM genes could provide novel targets for therapeutic interventions, particularly for those aimed at promoting phenotypic reversion, as already observed [25]. Finally, when OM genes’ coordinated expression exceeds a given threshold, the CP synchronizes with the GA, thus driving genome expression transition. Here, we will give an account of the mechanism by which this transition (initiated by the same OM genes in charge of homeostasis) happens.

The paper is organized as follows: Section 2 describes how, at the critical transition, OM genes dynamically modulate the synchronization of the CP and GA; this was conducted through the application of the Expression Flux Analysis (EFA: [15,16]) on time series genome expression data. Section 3 provides a potential mechanism for the synchronization between the CP and GA, and Section 4 gives a discussion of the biological basis of a potential unified SOC mechanism for the reversion of cancer cell-fate change, while Section 5 reports general conclusions.

## 2. Synchronization Dynamics between the CP and GA Drive Critical Cell-Fate Transitions through the Action of Oscillating-Mode Genes

In our previous works, we identified the variance in gene expression as an important metric for characterizing the dynamics of genome expression rearrangement. The degree of *nrmsf* (Normalized Root Mean Square Fluctuation) is the metric parameter for the self-organization of genome expression [15,16]. When, at each experimental time, gene expressions are sorted according to the value of *nrmsf*, from high to low, and arranged in groups (with the number of genes in each group *n* > 50), the Coherent Stochastic Behavior (CSB) becomes evident. In each group, the average expression quickly converges to a specific value as the size of the group becomes sufficiently large, according to the law of large numbers. Within those clusters, a particular set of genes displays a bimodal singular behavior that acts as a critical point (CP).

The main difference between c-SOC and our model of SOC is that, in the latter, whole-genome expression is coordinated by the critical set of genes (CP genes) behaving as the critical point (CP). In our model, whole-genome expression self-organizes into coexistent local critical states (distinct critical response domains), with the CP serving as a boundary between critical states (see the differences between a population of cells and a single cell in [13]). The presence of multiple local organizations of the system corresponds to a “chimera states” structure [26] and adds biological realism to c-SOC.

At the critical transition, an inversion of the bimodal singular behaviors at the CP occurs, (Figure 1; refer to updated descriptions of SOC methods in [Tsuchiya, M., et al., 2023] and more results in [13]), where the ON or OFF state change occurs within the CP, therefore determining the “collective” activation/inactivation of critical genes. The state change within the CP happens according to the change in critical behaviors, e.g., the disappearance of sandpile-type critical behaviors and switching bimodal behaviors (refer to [13]). Thus emerges the following pivotal question:
How does the genome avalanche (global expression change) at the critical transition occur in accordance with the state change within the CP?

The CP partially overlaps with the GA (see Figure 1C and Figure 2). In our study on HRG-Stimulated MCF-7 cells [15], we observed that the CP and GA synchronize at the critical transition (Figure 3; refer to the potential biophysical reason for the synchronization in Section 3). Noticeably, in control experiments, the EGF stimulus (that is not able to induce differentiation) does not provoke a critical transition in MCF-7 cells, and synchronization of the CP and GA does not happen (Figure 3B). On the contrary, in the case of HRG (efficient stimulus) at the critical transition, the synchronization between the CP and GA does occur (Figure 1). This synchronization induces a sequential change in the CP, which, in turn, determines a significant state change (genome avalanche: Figure 3B). The timing of the critical transition occurs at 15–20 min (ON to OFF for HRG-Stimulated MCF-7 cells). Furthermore, the timing of the critical transition coincides with that of the switching genome engine for HRG-MCF-7 cells (see Section 2.2).

**Figure 1 ijms-24-11603-f001:**
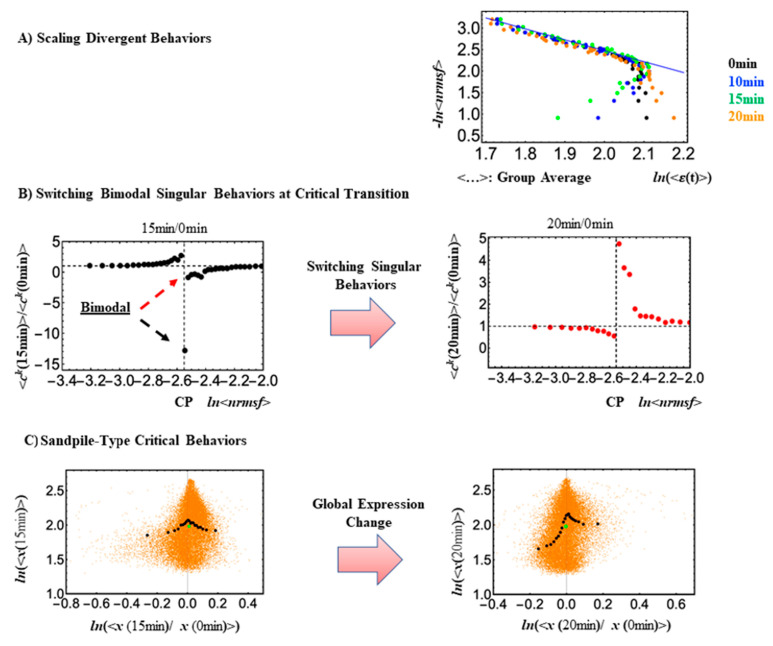
Critical Behaviors at CP in HRG-Stimulated MCF-7 cells. In both panels (**A**,**B**), gene expressions are sorted according to their *nrmsf* value, from high to low, and then grouped into 30 groups with an equal number of 742 elements. (**A**) Dots of different colors represent expression responses at different experimental time points, and the *x*- and *y*-axes are natural logs of (average) group expressions and corresponding *nrmsf* values, respectively. Low-variance genes *(ln*<*nrmsf*> ~−2.55), are located below the CP, and the scaling behavior is evident. Low-variance genes (subcritical state) behave collectively, whereas genes above the CP, i.e., genes with high variance (supercritical state), exhibit divergent behavior. (**B**) Along the metric parameter, genes around the CP region (group average; black dots: 0–15 min; red dots: 0–20 min) exhibit an inversion of bimodal critical behaviors (red dashed arrow for upregulation; black one for down-regulation) at 15–20 min; the *x*-axis here represents the ratio of CM values of the group. (**C**) Whole expressions are sorted and grouped based on the degree of temporal expression change (an expression group vector is denoted as ***x***(*t*_j_) at *t* = *t*_j_). A plot of expression temporal changes for each group in log-log space reveals a sandpile-type critical behavior (solid black dots: CM values of the group) at the critical point (the sumit of black dots), where the CP in terms of the ensemble (group) average (<…>) occurs at the near-zero change (*x* = 0; null expression change, i.e., the ratio of CM values is equal to one). The solid green dot represents the CM value of whole expression, i.e., genome attractor (GA).

Looking at Figure 1C, it is worth noting how small expression changes near the CP diverge to either up- or downregulating processes, consistent with a sandpile-type criticality. The CP exists nearby the genome attractor (GA: solid green cycle). At the critical transition, a change in the center of mass of the whole expression (solid green: GA), from positive at 0–15 min to a negative change at 0–20, denotes a change in the whole expression from up- to downregulation (Figure 2). Orange dots represent single mRNA expressions in the background between different time points. A detailed description of the methods for SOC analysis is provided in [15,16].

In Figure 3, it is worth noting that, in the absence of a critical transition, (EGF-Stimulated cells) the oscillatory behavior is absent. In fact, EGF (Epidermal Growth Factor) is unable to initiate the differentiation process, at odds with HRG (Heregulin) stimulus, which ignites a cascade of events resulting in cell differentiation.

### 2.1. Genome Avalanche to MCF-7 Cancer Genome: Spatio-Temporal Behavior of the Critical Transition

The spread of the differentiation stimulus across different chromosomes makes any portion of the CP belonging to a chromosome follow almost the same temporal expression response (see Figure 2.7 in [15]). At the critical transition, the average expression of each chromosome synchronizes with its CP-related genes. This implies that CSB can also be detected at the chromosome level, where the stochastic gene expression within a chromosome converges to its CM.

It is worth noting the existence of different levels of synchronization of chromosomes with the global CP (Figure 4A,B; see more in [15]). The clustering of synchronized and unsynchronized chromosomes (Figure 4B) shows how and where the critical transition spreads across the genome. The GA is located at the branching of synchronized chromosomes and the self-flux of the GA travels along two paths: one from the GA to chromosome 22, and the other from the GA to chromosome 17. Here, self-flux corresponds to the second-order time difference of the (group) average gene expression and represents an “effective force”. This force can be interpreted as a sort of “biophysical energy” transferred to the chromosome when its sign is positive and released from the chromosome when its sign is negative (see more in [16]). These paths reveal the spatio-temporal behavior of the critical transition with an amplified change in the GA (“energy transfer”) in the chromosome territories. Therefore, synchronization of the GA with the CP guides a genome-wide avalanche (domino effect) allowing a synchronized wave to spread through the chromosomes (Figure 4B).

### 2.2. Oscillating-Mode Genes Acting as Modulators of the CP and GA Synchronization: Determination of the Timing of Cell-Fate Change

The application of Principal Component Analysis (PCA) to temporal gene expressions [14] allows for the detection of the decoupling of a set of specific genes from the global set of genome expressions at the onset of transition. In PCA terms, this corresponds to the drastic increase in variance explained by minor (by construction independent from the main flux of variation) components. The presence of a strong, cell-type-specific gene expression invariant profile (the cell-type attractor; Figure 5) generates a first principal component (PC1) that explains a major part of the data set variance. The main orthogonal direction, with respect to PC1, is PC2 (second principal component), which corresponds to the main direction of motion around the cell-type attractor. Genes with maximal temporal variation around their equilibrium position drive PC2. We call these genes Oscillating-Mode (OM) genes. The continuous oscillations (mirrored by PC2 and other very minor components) around the ideal profile (PC1) are responsible for the homeostatic adaptation to minor environmental perturbations (small avalanches, in the SOC terminology). These oscillations reflect the apparent conundrum of sandpile SOC regulation, in which the attractor state is a critical state characterized by small avalanches functional to the stability of the gene expression profile when exposed to continuous small environmental perturbations [28,29].

The CP formalism, as described in Section 2, is coherent with the PCA approach in consideration of the fact that (local) supercritical genes correspond to the main drivers of minor components that, at CP, transmit their activation to the core (GA). The CP, determined by the metric parameter (*nrmsf*) of self-organization, serves as the organizing center of cell-fate change; its activation (or deactivation) makes local perturbation spread over the entire genome. It is worth noting that the specific signatures of SOC (refer to Figure 1) can be recognized in all of the above-described cases.

At the critical transition for HRG-Stimulated MCF-7 cells (15–20 min), PC2 genes are clearly activated (Figure 5), with a corresponding increase in variance explained by PC2. On the contrary, largely invariant genes have near-zero scores on minor components and correspond to the core of the sandpile. These genes constitute the CM of the system (genome attractor: GA) and are put into motion only when the transition is activated by a domino effect summing up the small avalanches. This process ends up in a new attractor state [14,23].

Therefore, we investigate how high PC2 score genes (|PC2| > 2: OM genes) dynamically modulate the synchronization of the CP and GP at the critical transition (threshold value stems from the fact that principal components are expressed in terms of z-scores, i.e., with zero mean and unit standard deviation). The temporal average of the flux network between the OMG–CP–GA network (OMG stands for OM genes) reveals that external flux outside of the OMG–CP–GA network is relatively small (Figure 6), which indicates that INcoming and OUTgoing fluxes in the network are almost balanced (i.e., forming a weakly non-equilibrium thermodynamic system) for MCF-7 cell response. Hence, we investigate fluctuations from the temporal average of the OMG–CP–GA network in order to uncover how OM genes temporally regulate the synchronization between the CP and GA. It is worth noting that the increase in the variance in minor components corresponding to OM gene hyper-activation happens with the same timing as for the genome engine switching.

As demonstrated in Figure 3A, the HRG-Stimulated MCF-7 cell response shows that the synchronization in self-flux (effective force) between the CP and GA occurs at the 10–30 min period and 2–4 h period windows. In the former case, small fluctuations in local critical-state attractors coherently switch the genome engine (critical-state attractor network; Figure 6) from suppression to enhancement (see Figure 7A). This non-linear behavior of small fluctuations plays a crucial role in the genome engine to switch it to the cell-fate-guiding genomic transition. On the other hand, for the temporal expression flux in the OMG–CP–GA network, Figure 7B demonstrates that OM genes emit large fluxes to both CP and GA at 15 min. However, the CP and GA actually receive relatively small fluxes from OM genes and a large flux (obtaining free energy or entropy) from the external flux, i.e., outside of the OMG–CP–GA network. These non-linear behaviors clearly demonstrate the broken mass-action law of the non-equilibrium process in the genome and send a large flux back to the OM genes. Similar INcoming and OUTgoing flux behaviors in the CP and GA exhibit synchronization of the CP and GA during that 20–30 min. Internal interaction fluxes are negligible this time, so that the lack of cyclic flux between the CP and GA eliminates anti-phase correlation behaviors. This behavior is reminiscent of the crucial role played by the microenvironment (external to the network) on the differentiation process.

Furthermore, the timing of switching in the OMG–CP–GA network coincides with that of the genome engine (Figure 7), where the OMGs suppress cyclic fluxes between the CP and OM genes and between the GA and OM genes at 15 min and 30 min, their cyclic fluxes enhanced to support the switching of the OMG–CP–GA network. The synchronization of the CP and GA also occurs in the 2–4 h time window. The erasure of the initial-state sandpile criticality (Figure 4D in [13]), which erases the initial memory of the CP genes, occurs after 3 h. These results are consistent with the timing of the biological commitment of differentiation at 15 min and late transcription activities at 3 h, leading to the determination of the differentiation state, coupled with the suppression of proliferation and stopping the genome boost through the ERK pathway (see more in [23,27]). Thus, we conclude that cell-fate change, primed at 15–20 min, occurs after 3 h in HRG-Stimulated MCF-7 cells.

## 3. Structural Bases of Genome Expression Dynamics

Here, we put forward a potential mechanism for the synchronization between the CP and GA. The Yoshikawa group [30,31,32,33,34] revealed that the higher-order structure of DNA, above the scale of several tens of kbps, undergoes a first-order phase transition between its elongated and compact states. Accompanied by various physio-chemical environmental changes, the free energy barrier on the first-order phase transition undergoes a drastic decrease and tends to exhibit noticeable fluctuations when approaching criticality. As Figure 1B shows, the bimodal singular behavior, nearby the CP, exhibits both activated and suppressed expression states, and its inversion at the critical transition suggests the formation of a phase segregation (bimodal domain of up- and downregulation) with a free energy barrier. When the free energy barrier is lowered, the dynamics of the CP synchronize to those of the GA to induce the cell-fate guiding critical transition.

The bimodal character of the free energy barrier between the CP and GA has a structural counterpart highlighted by a recent finding related to chromatin transitional dynamics, regarding the spatial interaction of two chromatin types. Silencing the Pericentromeric Associated Domains (PADs) that keep the chromatin in its collapsed heterochromatic state allows for the transcription of euchromatin domains [23]. Thus, understanding the higher-order structural change in genome-sized DNA in relation to the mechanism of control of several thousand genes is mandatory for comprehending gene expression controls in cell-fate change. Recently, such spatial interaction at the basis of chromatin dynamics was observed by Parmentier et al. [35], demonstrating how global chromatin decompaction, and the consequent stochastic gene activation, constitutes the initial step toward fate commitment in human hematopoietic cells.

In the case of MCF-7 cells perturbed by HRG, the global departure from gene expression identity (coming from the existence of GA) is maximal between 15 and 20 min. After that, the effect of the perturbation vanishes and the system comes back to its ‘business-as-usual’ behavior. By this time, it seems as though nothing relevant for the cell transition has happened. Nevertheless, if we wait for a much longer time, we will notice that this ‘shaking’ was not without consequences. The initial ‘wave of change’ dissipates after 30 min (the system goes back to its initial equilibrium gene expression profile), but the ‘memory’ of the initial perturbation remains for around 1–1.5 h, due to the action of FOS, a proto-oncogene of retroviral origin which plays a key role in both tumorigenesis and tumor suppression. In particular, FOS downregulates MAPK/ERK pathway, a cascade of subsequent protein activation that communicates signals from membrane receptors to the cell nucleus, causing tumor growth. ERK activation is abated by 3 h; after this time, the system is committed to leaving the neoplastic fate and entering the path leading to the terminally differentiated state, evident after one week.

Can we link these gene expression results to events happening at the chromatin organization level?

The huge size of the DNA molecule, compared to the size of a cell nucleus, forces this molecule to stay in an extremely collapsed compact state that must be selectively unfolded in order to open the way for a drastic change in gene expression. Figure 8 depicts the link between phenomenology (gene expression) and the reorganization of chromatin (change in number and size of Pericentromere-Associated Domains, PADs). The centromere of chromosomes is rich in so-called ‘satellite repeats’, which are very repetitive DNA sequences condensed as heterochromatin. This pericentromeric heterochromatin acts as a repressive environment in the nucleus. These regions are visible as ‘knots’ via microscopy and are called PADs. The changes in both PAD size and number correspond to a change in transcription activity through the folding/unfolding of chromatin, thus changing the accessibility of chromatin patches to gene expression. Control cells (ST, black squares), except from a few outliers, have a constant number of PADs, mainly concentrated in a narrow area range (between two and four squared micrometers). At the critical point (15 min after HRG, orange circles), the situation is completely different: the PADs are smaller and much more numerous, with a strong negative correlation between dimension and number, pointing to a dramatic unfolding of chromatin through the splitting of large PADs into smaller ones.

## 4. Discussion

Critical transitions in cancer have been observed in response to both internal and external cues. Indeed, cancer can be stimulated either to acquire new malignant traits or to revert to a non-invasive, physiological phenotype [25]. Destabilization of the interplay among cells and their microenvironment promotes a critical state transition—by analogy with those observed in physical systems—that can be driven by a number of specific cues to access new attractor states (i.e., phenotypic configurations). This reversion is associated with the modulation of a set of “critical genes”, which control a few numbers of key pathways (i.e., epithelial–mesenchymal transition [EMT], migration, invasiveness). The reversion of EMT should be viewed as an earlier and inescapable event during phenotypic reversion [36,37].

During such a reversion, cancer cells re-express several previously repressed genes, including those involved in cell plasticity (Oct4, SOX-2, and Nanog), while genes (PSEN1, Notch-1) and proteins (SNAI1, αSMA, N-cadherin) supporting epithelial–mesenchymal transition (EMT) are downregulated [38]. Overall, a general destabilization of the chromatin organization occurs, enabling cells to regain a pluripotent state, from which they can later be forced into a new, differentiated phenotype. Indeed, epigenetic changes associated with tumor reversion require several preliminary events before critical enzymatic pathways can be inhibited or enhanced to support the transition toward a new attractor. Specifically, chromatin should be “opened” and drastically remodeled, eventually through cytoskeleton (CSK) and nucleoskeleton (NSK) reshaping, in order to facilitate access to gene regulatory regions [39]. Both CSK and NSK are finely tuned transducers of mechanical and biophysical forces modulating gene expression. It is worth noting that some reverting factors can trigger tumor reversion in cancer cells by drastically rewiring the cytoskeletal architecture [40]. Furthermore, a number of oocyte/embryo components—including polymerase II—facilitate the rewiring of genome expression by speeding up the rate of transcription, promoting chromatin “de-condensation” so as to access previously quiescent segments of the genome [41], and, finally, enabling a “global reversal” of DNA methylation to make somatic nuclei closely resemble those of stem cells [42].

However, recovering a stem cell-like profile would be, in principle, incompatible with tumor reversion, since “uncontrolled plasticity”—as shown by induced pluripotent cells—can give rise to tumor development in a permissive milieu [43]. The attractor’s destabilization and subsequent “reprogramming” into a non-malignant phenotype could probably not rely solely on internal mechanisms, and may likely require the presence of tissue-dependent constraints that will ultimately drive the process toward a specific direction [44].

For this reason, the application of SOC analysis to time series genome expression data is crucial for identifying those critical transition points in which cells/tissues show high sensitivity to even weak control actors [45]. It is precisely at these points that the system can be effectively “displaced” from its quasi-stable state, and then “driven” towards different/alternative cell-fate commitments.

## 5. Conclusions

Through this review paper, focusing on MCF-7 cancer cells as an example, we described the putative genomic mechanism of cell-fate change [15] that has been supported by recent chromatin experimentation [23], as well as multivariate statistical analysis [14].

Our studies [13,15] revealed how the complex and dynamically evolving molecular networks found in biological systems can give rise to globally coherent orchestrated responses. We have demonstrated that the identified critical point (CP) in gene expressions serves as the organizing center of cell-fate change, and its activation or deactivation causes the spread of local perturbation over the genome, affecting the genome attractor (GA). Cell-fate change asks for massive control of whole-genome expression at the critical transition through the synchronization of the CP and GA. Expression Flux Analysis (EFA: 15,16]) demonstrates that this synchronization happens through the modulation of Oscillating-Mode (OM) genes. These genes are normally in charge of genome homeostasis, to keep gene expression levels on pace with microenvironment fluctuations. The critical transition occurs to provoke cell-fate change when the perturbation entity on OM genes exceeds a certain threshold. This model can be viewed as a sort of ‘domino-effect’, in which small avalanches, initially confined in the periphery of the system, invade the core (GA) of the system, putting ‘in motion’ the genes normally preserved by microenvironment vagaries. Right before the critical transition, large-expression flux is either emitted to or received from the CP. This is a signature that oscillation exceeds the threshold value, allowing CP–GA synchronization, thus acquiring the possibility of driving the state transition.

The SOC analysis provides further understanding of the critical transition spatio-temporal behavior within the cancer genome. The CP genes spread over all chromosomes, where their averaging behavior of the CP portion exhibits similar dynamic behaviors to the overall CP dynamics. It is worth noting that there exist both unsynchronized and synchronized chromosomes; a CM of the gene expression in a chromosome synchronizes dynamically to that of the CP over experimental time points (high temporal Pearson correlation). At the critical transition, a change in the GA spreads over the synchronized chromosomes via the amplification of the temporal change in the GA. This is consistent with the spatio-temporal behavior of the critical transition, demonstrating an avalanching process which exhibits domino effect in the cancer genome.

The proposed mechanism represents a unifying step toward a time-evolutional transition theory of biological regulations, especially for the development of possible control strategies for cancer cell fates. The suggested SOC model has a material counterpart made evident by the temporal correlation between the transition in gene expression and chromatin folding/unfolding dynamics that make the regulation of large ensembles of genes physically feasible.

The search for effective ways to induce cancer reversion is probably the most ambitious goal to which the results described in this review could be advantageously applied. In any case, the proposed approach is not a singular exploit. Many statistical mechanics approaches to gene expression regulation have recently flourished, and they are essentially convergent with our SOC model (see, for example, [35,46,47]). Of particular interest are the multiscale approaches that combine the gene expression cell-based dynamics toward attractor states and the dynamics of cell–cell interactions, giving rise to a complex tissue architecture [48]. All in all, we are convinced that cell biology is entering into a new era, which will foster a fruitful interaction between the gene-centric perspective of recent decades with physically oriented integrative approaches to biological systems considered according to their proper nature of interaction networks.

## Figures and Tables

**Figure 2 ijms-24-11603-f002:**
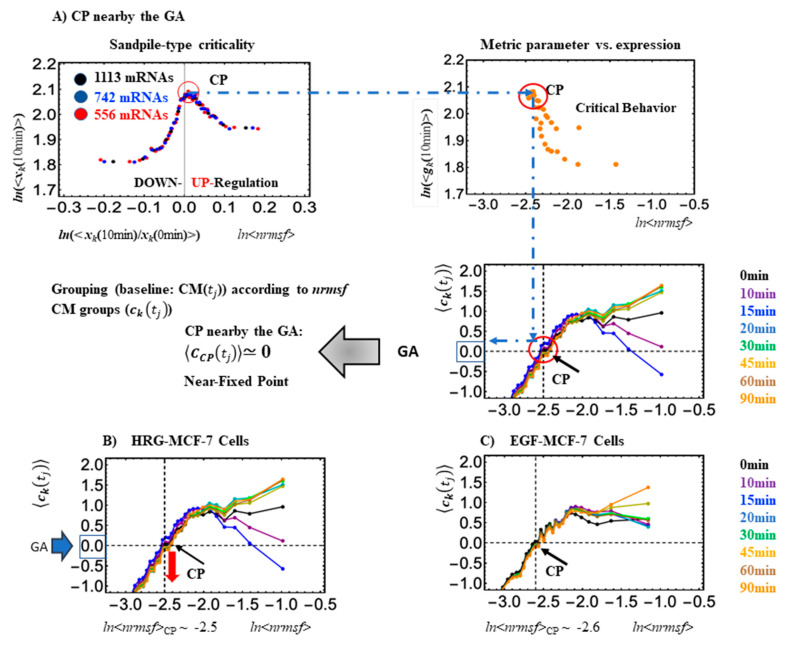
CP and GA. (**A**) Top left panel. In HRG-Stimulated MCF-7 cells, the same kind of sandpile-type critical behavior is displayed by different size (*n*) groupings (Here, the *k*th group vector is denoted as ***x***_k_(*t*_j_), *k* = 1, 2, …, *K*; *K* = 20 groups corresponding to black dots, 30—blue, and 40—red; groups contain 1113, 742, and 556 mRNA species, respectively). This suggests the existence of scaling behaviors (i.e., renormalization) due to CSB. Top right panel: Based on groupings by the degree of *nrmsf*, the logarithm plot of the average *nrmsf*, *ln*<*nrmsf*>, vs. CM of the group, $ln\langle g_{k}\left(10\;\min\right)\rangle$, at t = 10 min shows that the summit in sandpile critical behavior corresponds to *ln*<*nrmsf*>_CP_ ~−2.5 to −2.6 (<…> represents the ensemble average). Bottom right panel: CM grouping (***c***_k_(*t*_j_)) according to the degree of *nrmsf* reveals that the CP can be considered a nearly fixed point (time independent) and exists in the vicinity of the genome attractor GA (see more in Figure 2 in [13]). (**B**) HRG- and (**C**) EGF-Stimulated MCF-7 cancer cells. These plots show that the CP exists near the GA. The different colors of the dots represent responses at different experimental times.

**Figure 3 ijms-24-11603-f003:**
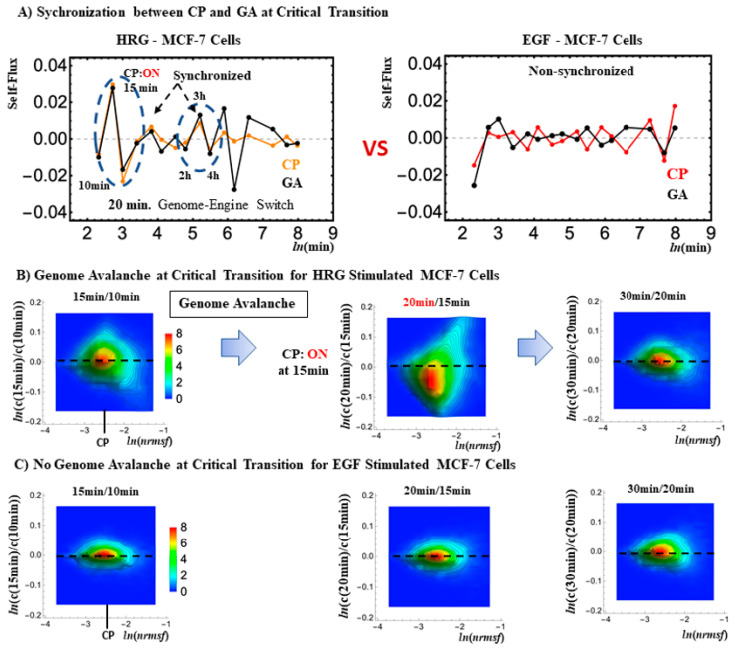
Synchronization of the CP and GA at Critical Transition with global avalanches. Panel (**A**) (left): In HRG-Stimulated MCF-7 cells, noticeable synchronization of the CP and GA occurs at 10–30 min and 2–4 h. As for the 10–30 min period, global avalanches occur: at 15–20 min, the critical transition happens with ON–OFF switched in the CP (CP: ON at 15 min; CP: OFF at 20 min), where the CP timing of ON–OFF coincides with that of the upregulated and downregulated whole expression, respectively (panel (**B**)). As for 2–4 h, cell-fate change occurs after 3 h [23,27] with CP being ON. Panel (**C**): In EGF-Stimulated MCF-7 cells (cell proliferation without cell-fate change [17,18]), synchronization of the CP and GA does not occur, and, thus, no global avalanche happens within the same time window. In panel (**A**), CP responses are amplified seven times.

**Figure 4 ijms-24-11603-f004:**
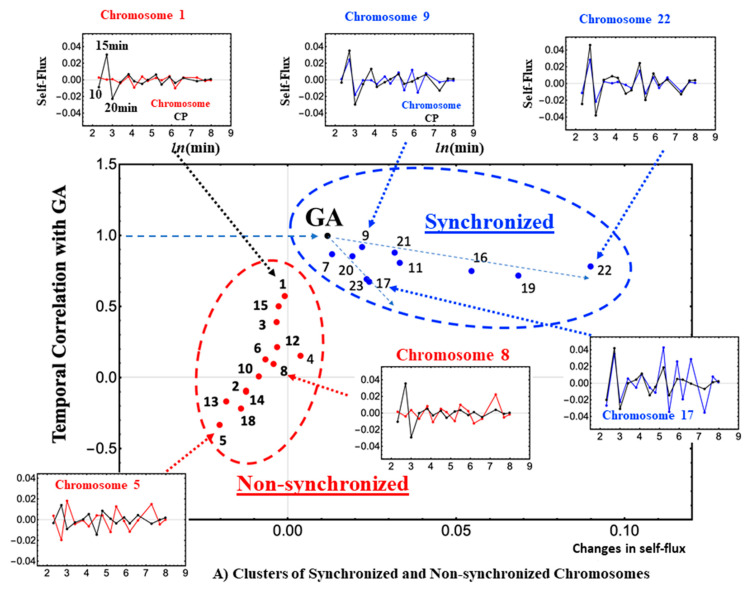
(**A**). At the critical transition (10–15–20 min), chromosomes are either synchronized or unsynchronized with the CP dynamics (black lines in small plots). Small plots show examples of synchronized (9, 17, 22: blue lines) and unsynchronized (1, 5, 8: red lines) chromosomes’ self-fluxes with the CP dynamics (*x*-axis: In the background plot (large plot), the *x*-ax is represents change in self-flux; *y*-axis: self-flux). The numerical values of the self-flux of chromosome 9 and 17 CPs are magnified by three and five times, respectively, and the same holds for chromosomes 1, 5, and 17. In the background plot (large plot), the *x*-axis represents change in self flux, and the *y*-axis shows a temporal Pearson correlation in the dynamics of the GA. (**B**) The GA is located at the edge of two paths (chromosomes 9 and 7), where the self-flux of the GA is amplified through them (i.e., genome avalanche). This reveals the spatio-temporal dynamics of the critical transition through the synchronized chromosomes. The *x*- and *y*-axes use the same scale of (**A**).

**Figure 5 ijms-24-11603-f005:**
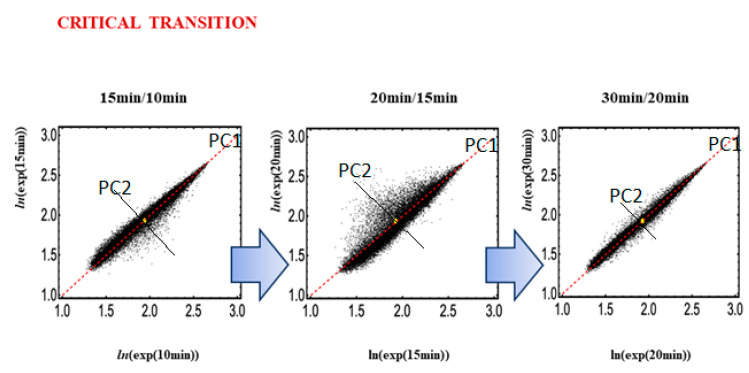
For HRG-Stimulated MCF-7 cells, Pearson’s correlation between gene expression profiles at different time points reveals the occurrence of the critical transition at 15–20 min, around the CM (gene expressions: back solid dots; CM of whole expression: yellow dot), corresponding to the decrease in Pearson r and, visually, to the increase in the scattering of points from the diagonal identity line. Axes (black lines) for the first (PC1) and second (PC2) principal components are shown. At the critical transition, the genes that are most loaded on PC2 (which are oscillating around the PC1 invariant profile) exhibit maximum departure from the invariant profile, corresponding to a net increase in the proportion of variance explained by the second component.

**Figure 6 ijms-24-11603-f006:**
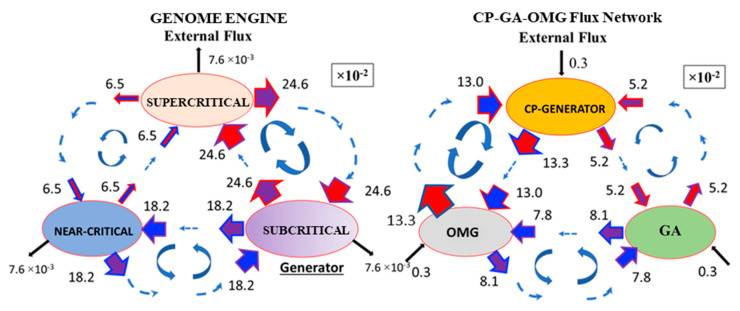
Temporal Average of the Flux Network for HRG-Stimulated CP, GA, and OM Genes. MCF-7 cancer cell response; left panel: The diagram demonstrates the genome engine mechanism (see more in [15]). Subcritical state interactions generate a dominant flux through which to establish the genome engine mechanism. The subcritical state acts as a “large piston” for small movements (low-variance expression), while the supercritical state behaves as a “small piston” for large movements (high-variance expression). A sort of “ignition switch” (critical point) connects the different states through a dominant cyclic state flux (which we can equate to a ‘camshaft’), resulting in the anti-phase dynamics of the two pistons. Numerical values show average between-state expression fluxes based on a 10^−2^ scale. Right Panel: Temporal average of the flux network for OMGs (Oscillating-Mode genes: |PC2| > 2.0), the CP, and the GA. The external flux for both the genome engine and the CP–GA–OMG network is relatively small, such that INcoming and OUTgoing fluxes are almost balanced (i.e., weakly non-equilibrium system).

**Figure 7 ijms-24-11603-f007:**
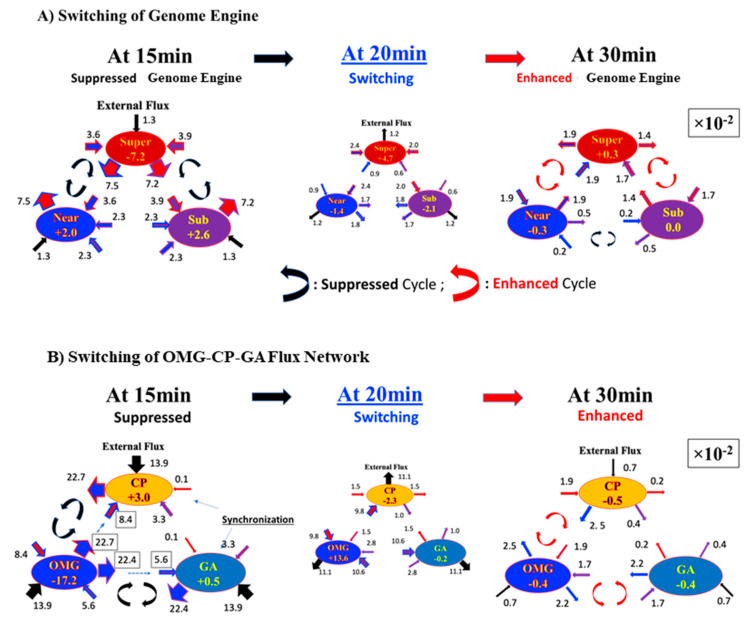
Switching of the OMG–CP–GA flux network coincides with switching of the genome engine for MCF-7 cancer cells. (**A**) Switching of the genome engine and (**B**) switching of the OMG–CP–GA flux network. The flux network among OM genes (Oscillating-Mode genes: |PC2| > 2.0), the CP, and the GA, where the OM genes modulate/generate the synchronization of the CP and GA. Switching of the flux network occurs at the time when the genome engine is switched, (**A**). Furthermore, the flux network reveals synchronization between the CP and GA, which demonstrates similar IN and OUT flux behaviors in the CP and GA, while exhibiting non-linear behaviors (broken mass-action law) and switching of the OMG–CP–GA network coinciding with those of the genome engine switch. Flux represented by arrows with different colors are the same as Figure 6.

**Figure 8 ijms-24-11603-f008:**
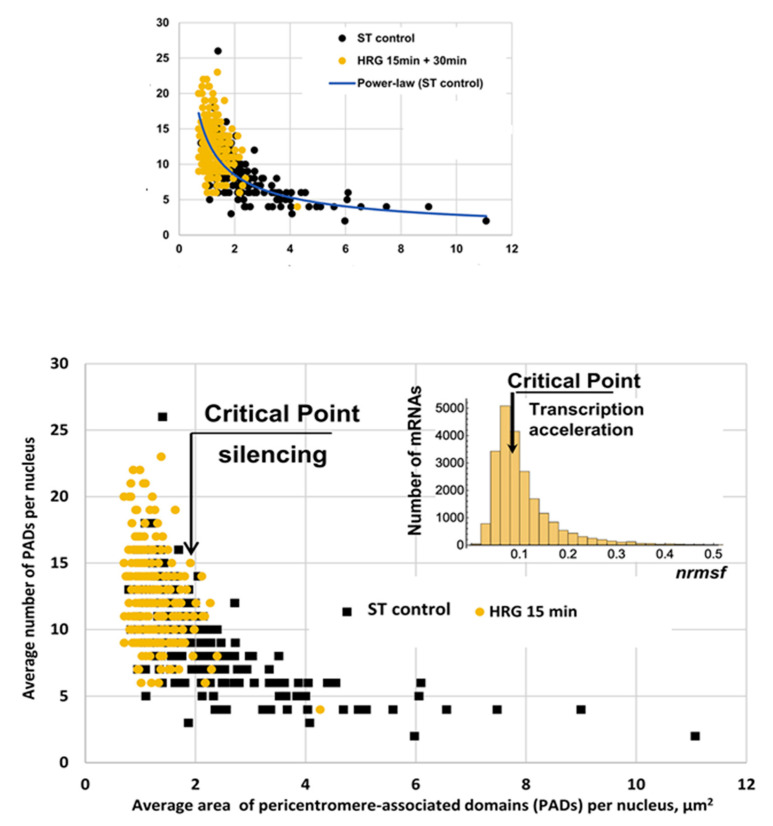
The *x*-axis represents the size (area) of PADs, and the *y*-axis represents their per-nucleus average numbers. Black dots correspond to a non-critical condition, while yellow dots refer to HRG stimulus at the transition point (15 min). The distribution of size and number of PADs follows an approximate 1/*f* scaling (hyperbolic negative relation between size and number, top panel, with a scaling exponent equal to -0.67 and an R^2^ = 0.56) that is a hallmark of the dynamic fusion splitting of PADs, typical of the ‘edge-of-chaos’ condition. The inset of the figure (bottom panel) reports the normalized standard deviation of expression (*nrmsf*) for the 22,277 genes (*x*-axis) together with the total number of messenger mRNAs (*y*-axis), which is a crude but effective index of transcription activity. It is immediate to note both the global similarity of the two distributions and the corresponding position of the 15 min critical point. Both viewpoints are consistent in indicating the commitment of differentiation at 15 min, suggesting the unfolding of chromatin as the main driver of the process. This unfolding is registered by the increase in number and shrinking in size of PADs, signaling a drastic remodeling of chromatin at the critical transition and allowing for a global change in genome expression [23].

## Data Availability

Microarray data of the activation of ErbB receptor ligands in MCF-7 human breast cancer cells stimulated by EGF and HRG; Gene Expression Omnibus (GEO) ID: GSE13009.

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
