# Peer review of "Synchronization between Attractors: Genomic Mechanism of Cell-Fate Change"

_ijms, 2023, doi:10.3390/ijms241411603_

Round 1

Reviewer 1 Report

In this manuscript, the authors reviewed the genome activities in HRG stimulated MCF7 cell by using the dynamic criticality approach. Identifying the presence of oscillating mode genes helps the current understanding of cell-fate transition. The work has shown the potential genome therapeutic innovation in the future.

The manuscript is well written. The arguments are clear and significant. Publication of this manuscript is recommended.

Two minor editorial suggestions are as follows.

(1) In Fig. 2 B), its x-axis label overlaps with the scale.

(2) The authors introduced multiple abbreviations and variables in the manuscript. It may be more convenient for a wide range of readers to go over the content if all the abbreviations to shorten terminologies and symbols to represent variables can be summarized in one table (besides their definitions given in the body of the manuscript).

Author Response

In this manuscript, the authors reviewed the genome activities in HRG stimulated MCF7 cell by using the dynamic criticality approach. Identifying the presence of oscillating mode genes helps the current understanding of cell-fate transition. The work has shown the potential genome therapeutic innovation in the future.

The manuscript is well written. The arguments are clear and significant. Publication of this manuscript is recommended.

Two minor editorial suggestions are as follows.

(1) In Fig. 2 B), its x-axis label overlaps with the scale.

(2) The authors introduced multiple abbreviations and variables in the manuscript. It may be more convenient for a wide range of readers to go over the content if all the abbreviations to shorten terminologies and symbols to represent variables can be summarized in one table (besides their definitions given in the body of the manuscript).

We wish to thank the reviewer for the appreciation of our work and followed his/her suggestions in the revised version of the manuscript

Reviewer 2 Report

I consider the scientific topic addressed by the authors interesting.
However, there is a significant resemblance in certain components of this manuscript to their previously published book chapter (Tsuchiya, M. et al. 2022. A Unified Genomic Mechanism of Cell-Fate Change. In: Nuclear, Chromosomal, and Genomic Architecture in Biology and Medicine. Results and Problems in Cell Differentiation, vol 70. Springer).
Already the second part of the manuscript title “Genomic Mechanism of Cell Fate Change” strongly resembles the chapter name “A Unified Genomic Mechanism of Cell-Fate Change”.
In continuation, the submitted manuscript contains a number of images which are an exact duplication of the images presented in their book chapter (or with negligible changes), without any indication that a permission to reuse the content has been obtained from Springer (e.g. Fig 4B and C., Figure 5).  Some of the other Figures are only a minor adaptation from their other work, e.g. Tsuchiya et al. Int. J. Mol. Sci. 2020, etc.

Therefore, even if the authors decide to use this extent of their previous works as a base for this review paper, I think that the schematics and visualization throughout the paper needs to be reassessed and presented in a more original and less repetitive manner.

Author Response

Rev.2

I consider the scientific topic addressed by the authors interesting.
However, there is a significant resemblance in certain components of this manuscript to their previously published book chapter (Tsuchiya, M. et al. 2022. A Unified Genomic Mechanism of Cell-Fate Change. In: Nuclear, Chromosomal, and Genomic Architecture in Biology and Medicine. Results and Problems in Cell Differentiation, vol 70. Springer).
Already the second part of the manuscript title “Genomic Mechanism of Cell Fate Change” strongly resembles the chapter name “A Unified Genomic Mechanism of Cell-Fate Change”.
In continuation, the submitted manuscript contains a number of images which are an exact duplication of the images presented in their book chapter (or with negligible changes), without any indication that a permission to reuse the content has been obtained from Springer (e.g. Fig 4B and C., Figure 5).  Some of the other Figures are only a minor adaptation from their other work, e.g. Tsuchiya et al. Int. J. Mol. Sci. 2020, etc.

Therefore, even if the authors decide to use this extent of their previous works as a base for this review paper, I think that the schematics and visualization throughout the paper needs to be reassessed and presented in a more original and less repetitive manner.

We totally agree with Rev.2 and for this reason (as can be appreciated by the extensive changes of the actual version with respect to the previous one) we rephrased the parts of the manuscript resembling the previously published chapter and changed the relative figures. We operated the same ‘rephrasing and clarification’ procedure (that I hope was effective) across the entire manuscript so to better clarify the general message. 

Round 2

Reviewer 2 Report

The authors have made a number of changes throughout the manuscript. I think the present form of the manuscript could be accepted for publication.